# Socio-Emotional Outcomes of Child Care Participation: Results of a Four-Year Longitudinal Cohort Study

**DOI:** 10.3390/children12111463

**Published:** 2025-10-28

**Authors:** Mila Kingsbury, Leanne Findlay

**Affiliations:** Health Analysis and Modelling Division, Statistics Canada, Ottawa, ON K1N 8J8, Canada; leanne.findlay@statcan.gc.ca

**Keywords:** child care, longitudinal studies, socio-emotional development

## Abstract

**Highlights:**

**What are the main findings?**

**What is the implication of the main finding?**

**Abstract:**

Background/Objectives: Despite a wealth of research on potential socio-emotional outcomes of child care participation, results have been mixed, likely reflecting between-study differences in confounding variables assessed. Moreover, there is a need for updated studies using Canadian population data. The purpose of this study was to assess associations between child care participation between the ages of 1 and 5 years and socio-emotional outcomes four years later, accounting for a wide range of potential confounding variables. Methods: This study uses data from 8929 children ages 1–5 who participated in the 2019 Canadian Health Survey on Children and Youth, for whom follow-up data were available from the 2023 collection cycle. Parents reported on their use of child care for the target child in 2019; children’s socio-emotional functioning was assessed via a parent report in 2023. Associations between child care participation and outcomes were assessed using logistic and linear regression, adjusting for confounding factors including child age and gender, low family income, parental education, parental marital status, positive parenting, and education disruptions due to the COVID-19 pandemic. Results: Child care participation among 1–5 year olds was associated with higher scores for symptoms of attention-deficit hyperactivity disorder (ADHD), oppositional defiant disorder (ODD), and internalizing problems but a lower likelihood of having functional difficulty making friends four years later. Differences in these associations were noted based on the type of child care attended and family income, with centre-based care showing the most robust associations with ODD and internalizing symptoms, particularly among those living above the low-income threshold. Conclusions: These results highlight important contextual differences in associations between child care participation in early childhood and socio-emotional outcomes four years later. Associations with socio-emotional outcomes persisted after adjusting for important contextual factors, including positive parenting and COVID-related school disruptions.

## 1. Introduction

Decades of research have been devoted to the comparison of developmental outcomes between children who attend non-parental child care and those who are exclusively cared for by parents [1]. However, findings regarding the long-term associations between child care participation and socio-emotional outcomes have been mixed, due in part to heterogeneity between studies in terms of the consideration of confounding factors, as well as a distinction between studying associations based on targeted intervention studies in the United States (e.g., the Perry Preschool program [2] and the Abecedarian project [3,4]) as opposed to child care available to the general population.

Many studies have demonstrated that child care participation is associated with improved social, cognitive, and academic outcomes in early and middle childhood [5,6,7], with some studies suggesting educational and economic benefits lasting into adulthood [8]. However, other studies have found inconsistent long-term associations, particularly for non-cognitive skills [9]. A 2018 meta-analysis of studies of universal child care programmes reported that about one third of all studies report a positive association between child care participation and children’s outcomes (e.g., cognitive and social skills, school performance); 16% reported negative associations with these outcomes; and the remaining studies found no significant associations [10]. In terms of potential negative effects, child care participation in early childhood has been associated with more behaviour problems at kindergarten entry [11], with some studies suggesting a fade-out effect as children move through the later elementary grades [12]. A recent Canadian study using data from the National Longitudinal Survey on Children and Youth (NLSCY; 1994–2009) suggested that child care attendance between ages 3 and 5 was associated with higher levels of hyperactivity/inattention into the teen years [13].

Despite the wealth of research on associations between child care participation and child outcomes, there is a need for updated findings based on more recent longitudinal Canadian data. The most recent available findings come from the NLSCY, examining outcomes of child care participation during the 1990s and early 2000s. For example, Romano, Kohen, and Findlay [14] found that participation in child care was associated with greater physical aggression, and low child care stability was linked to greater hyperactivity–inattention, internalizing behaviour, and prosocial behaviours. However, parent-reported indicators of child care quality had an important interaction effect: children in home-based care characterized by high process quality (i.e., with warm, nurturing caregivers and appropriately stimulating experiences) were more likely to be prosocial, and children from low-income families but in high process quality care did not display greater aggression or internalizing behaviour. These results suggest that the quality of child care matters for preschool behavioural outcomes, even after controlling for socio-demographic factors, although child care process quality is difficult to measure and often not possible to explore as a moderator in research using survey data [15].

Additionally, many previous studies fail to consider the influence of confounding variables, including selection-related factors (e.g., family income, parental education), which may be associated with the quality of child care, as wealthier parents may have access to higher quality care [16,17], the amount of time spent in child care [12,18], as well as contextual factors including the family environment [19] and children’s other social experiences [20], which are likely to influence their socio-emotional and behavioural development in tandem with child care experiences. Parental sensitivity is highlighted as a key influence on child socio-emotional development, over and above the impact of child care experiences [19,21]. Moreover, parental sensitivity may differ between children who do and do not attend non-parental child care [1,21], highlighting the importance of examining this factor in studies of the associations between child care experiences and socio-emotional outcomes. Some previous work has suggested that child care experiences and parenting may interact—for example, one study suggested that time spent in child care was only associated with externalizing symptoms for children whose mothers exhibited low sensitivity [22]. Effects of child care participation have also been found to differ between socio-demographic groups—for example, with improved outcomes most notable among children living in low-income families [1,23]—and by gender, with some studies reporting that boys may be more sensitive to the effects of child care [24,25].

Taking children’s later school experiences into account may also be important to consider, especially in the context of the COVID-19 pandemic when school closures impacted many children’s experiences. Research indeed suggests that the pandemic and associated school closures were associated with adverse mental health symptoms, including distress, anxiety, and depression among children [26]. In a comparative cross-sectional study of preschool-aged children, those of preschool age during the pandemic showed lower personal–social skills than those who were assessed prior to March 2020 [27]. However, not all children were equally impacted. Families with young children were likely to have had a range of experiences with school closures and online learning, depending on their particular arrangements and circumstances. Some research suggests that experiencing school closures and remote schooling had small but negative effects on children’s mental health [28], making school experiences important contextual factors to consider when examining the longer-term outcomes of child care participation in the context of the COVID-19 pandemic.

### The Present Study

The aim of the present study is to assess associations between child care participation in early childhood and socio-emotional outcomes at school age, using recent longitudinal Canadian data while adjusting for key contextual factors. Specifically, this report aims to answer the following research questions:(1)Is participation in child care associated with longer term child socio-emotional outcomes four years later (i.e., at school age)? Do associations differ for children in different types of care and time spent in care?(2)Do associations between child care participation and socio-emotional outcomes remain significant after adjusting for selection factors such as family socio-economic status and parental education? Do such selection factors moderate associations between child care participation and child outcomes?(3)Are the associations between child care participation and socio-emotional outcomes moderated by children’s concurrent and later experiences, including parenting behaviours and educational experiences during the COVID-19 pandemic?

## 2. Materials and Methods

### 2.1. Data Source

Data were drawn from the 2019 and 2023 waves of the Canadian Health Survey on Children and Youth (CHSCY), a nationally representative longitudinal survey collecting information on issues affecting the physical and mental health of Canadian children. The 2019 sample covered the population ages 1 to 17, with the exception of children and youth living on First Nation reserves and other Indigenous settlements in the provinces, those living in foster homes, and the institutionalized population (covering an estimated 98% of the total population in the provinces and 96% in the territories). The longitudinal follow-up (2023) included a sample of those who responded to the 2019 CHSCY (retention rate: 54.2%), excluding those living in the three territories.

The analytical sample for the present study included N = 8929 children ages 1–5 years old in 2019 for whom follow-up data were available in 2023. Questionnaires were administered to the person most knowledgeable about the selected child (in over 96% of cases this was a birth parent; hereafter referred to as ‘parent’).

### 2.2. Measures

Child care participation. Children’s participation in child care was reported in 2019 by the parent (‘Do you currently use regular child care for this child?’; ‘yes/no’). Parents additionally reported on the type of child care (categorized for analysis as centre-based, home-based, or before-/after-school) and number of hours per week spent in child care in the past week (categorized for analysis as 0 h, 1–34 h, 35 h, or more).

Socio-emotional outcomes. Children’s behavioural and emotional functioning in 2023 was assessed using two measures. Items from the Washington Group/UNICEF Child Functioning Module [29] were used to assess functional difficulties in three areas: anxiety, depression, and social functioning. Parents reported on the frequency of children’s specific symptoms of anxiety (‘how often does [the child] seem very anxious, nervous, or worried’; ‘daily’, ‘weekly’, ‘monthly’, ‘a few times a year’, or ‘never’) and depression (‘how often does [the child] seem very sad or depressed’; ‘daily’, ‘weekly’, ‘monthly’, ‘a few times a year’, or ‘never’). Children were considered to have a functional difficulty in these domains if their parent reported that they experienced at least one of these symptoms ‘daily’. Children’s social functioning was assessed with the item: ‘does [the child] have difficulty making friends’, with responses: ‘no difficulty’, ‘some difficulty’, ‘a lot of difficulty’, or ‘cannot do at all’. Children were considered to have a social functioning difficulty if their parent selected ‘a lot of difficulty’ or ‘cannot do at all’ [30].

Parents additionally completed the Emotional–Behavioural Scale, initially developed for the Ontario Child Health Study (OCHS-EBS) [31]. The OCHS-EBS assessed children’s behavioural and emotional symptoms across three domains: attention-deficit hyperactivity (ADHD; 6 items, e.g., ‘distractible, has trouble sticking to any activity’; α = 0.85), internalizing symptoms (8 items, e.g., ‘anxious or on edge’; α = 0.83), and oppositional defiance (ODD; 6 items, e.g., ‘argues a lot with adults’; α = 0.82). Parents rated how true each item was of their child on a 3-point scale (‘never or not true’; ‘sometimes or somewhat true’; or ‘often or very true’). Item scores on each scale were summed to create total scale scores (scale ranges displayed in Table 1), with higher scores indicating higher symptom levels.

Baseline socio-emotional adjustment. To account for children’s adjustment at baseline, parents’ reports of their child’s general mental health (‘In general, how is this child’s mental health?’; ‘Excellent’, ‘Very good’, ‘good’, ‘fair’, or ‘poor’) were considered. This measure was dichotomized for analyses, comparing those with ‘very good’ or ‘excellent’ mental health to those with ‘good’, ‘fair’, or ‘poor’ mental health.

Covariates and contextual factors. The child’s age (in years, based on date of birth) was considered as a covariate. The child’s gender was reported by the parent in 2019 (‘male’, ‘female’, or ‘gender diverse’). Due to the low number of gender diverse children, analyses involving gender were restricted to boys (gender male) and girls (gender female). Parents reported their current marital status in 2019 and 2023; for the present study, these were dichotomized as married or living with a common-law partner versus widowed, divorced, separated, or never married at each time point. Change in marital status from 2019 to 2023 was then derived (new separation, new partnership, or no change). Respondents reported their highest level of educational attainment, categorized as a high school diploma or less, a certificate or diploma below the bachelor’s level, or a bachelor’s degree and beyond. Low family income was derived based on the ratio of the parent-reported household income to the corresponding low-income measure threshold for a family of the same size in 2019 [32].

Positive parenting was assessed using a 5-item measure initially developed for use in the National Longitudinal Survey of Children and Youth in 1994. Items (e.g., ‘how often do you and this child laugh together’) were rated on a 5-point scale (‘never’; ’about once a week or less’; ‘a few times a week’; ‘one or two times a day’; or ‘many times each day’), and item scores were summed to create a total positive parenting score (α = 0.75).

COVID-related education disruptions were assessed using two metrics. For children who had been enrolled in school any time between March 2020 and December 2022 (92% of children age 1–5 in 2019), parents were asked ‘overall, how was this child’s school learning impacted by the COVID-19 pandemic’, with the following response options: ‘improved a lot’; ‘improved a little’; ‘no impact’, ‘worsened a little’; or ‘worsened a lot’. This variable was categorized for analyses as follows: positive impact (improved a lot or a little); no impact; negative impact (worsened a lot or a little); or was not enrolled in school between March 2020 and December 2022. A second battery of questions was asked to parents of children who attended school at least partially online at any time between March 2020 and December 2022 (71% of children). Parents were asked whether their child had experienced any of 10 specific difficulties related to online learning (e.g., difficulty staying engaged; difficulty accessing learning supports) or any other unlisted difficulty. A summative variable was created, ‘difficulty with online learning’, comparing children who experienced at least one difficulty to those who reported no difficulties. A third category represented those who did not engage in online learning during this time.

### 2.3. Statistical Analysis

All analyses were conducted in SAS 9.4 using longitudinal survey weights and bootstrap weights with 1000 resamples. Weighted proportions and 95% confidence intervals were calculated for all variables of interest. Associations between child care participation (and child care type) and socio-emotional outcomes were assessed using logistic (functional difficulties) and linear (OCHS-EBS subscale scores) regression analyses. A second set of regression analyses adjusted for socio-demographic covariates, baseline mental health, positive parenting, and COVID-related learning disruptions. Interactions between child care participation and variables of interest (e.g., low income, positive parenting) were assessed by adding interaction terms to regression models in a third set of analyses. Where evidence for an interaction was found, stratified regression models were fit to examine associations between child care participation and the effect modifier.

Missing data analysis suggested that fewer than 1% of cases had missing data at baseline. Therefore, missing data were handled using analysis-wise deletion. Attrition analyses compared baseline socio-demographic variables between those who did and did not respond to the survey at time 2 using chi-square tests of independence. Children who did not participate in the 2023 follow-up were less likely to attend child care (53% vs. 60%, Χ^2^ = 55.3, *p* < 0.001), more likely to live in low-income households (34% vs. 24%, Χ^2^ = 188.9, *p* < 0.001), and less likely to have very good or excellent mental health at baseline (92% vs. 95%, Χ^2^ = 30.4, *p* < 0.001). Of note, the longitudinal survey weights are designed to adjust for nonresponse, providing more unbiased estimates.

## 3. Results

Descriptive characteristics are presented in Table 1. Approximately 60% of parents reported that they used regular child care for their child age 1–5 in 2019. In the past week, 31% of children had attended child care for fewer than 35 h per week, and 27% had attended for 35 or more hours. Of the children in child care, approximately half were enrolled in centre-based care, one third in home-based child care, and the remaining one sixth in before- or after-school care.

**Table 1 children-12-01463-t001:** Descriptive statistics, Canadian children ages 1–5 in 2019.

	%		95% CI	
Child gender, parent-reported, 2019 ^1^				
Boy	51.41		50.70	52.13
Girl	48.59		47.87	49.30
Child age, 2019				
1 year	18.6		17.6	19.7
2 years	19.9		18.8	21.0
3 years	19.9		18.8	21.0
4 years	20.4		19.3	21.5
5 years	21.2		20.0	22.3
Low family income, 2019	27.60		26.29	28.90
Single parent, 2019	12.72		11.62	13.82
Parent highest education, 2019				
High school or less	17.15		16.05	18.24
Certificate below bachelor’s level	34.18		32.81	35.55
Bachelor’s degree or beyond	48.67		47.25	50.09
Regular child care use, 2019				
No child care	39.91		38.55	41.28
Home-based child care	21.16		20.00	22.31
Centre-based child care	31.14		29.85	32.43
Before-/after-school child care	7.79		6.94	8.64
Time spent in child care in the past week, 2019				
0 h	41.35		39.97	42.74
<35 h	31.19		29.79	32.60
35 h or more	27.45		26.23	28.68
Parent-rated general mental health, 2019				
Good, fair, or poor	6.5		5.7	7.2
Very good or excellent	93.6		92.8	94.3
COVID’s effect on learning (2023)				
Positive effect	9.08		8.27	9.90
No effect	38.08		36.76	39.39
Negative effect	44.88		43.54	46.21
N/A (not enrolled in school between Mar 2020 and Dec 2022)	7.97		7.14	8.79
Difficulties with online learning during COVID-19 (2023)				
Yes	48.21		46.84	49.58
No	23.11		21.88	24.33
N/A (no online learning between Mar 2020 and Dec 2022)	28.69		27.45	29.93
Marital status change, 2019–2023				
New separation	6.10		5.36	6.84
New union (married or common-law)	2.93		2.36	3.49
	Range	Mean	95% CI	
Positive parenting, 2019	0–20	16.47	16.39	16.54
Functional difficulties, 2023		%	95% CI	
Making friends		3.16	2.66	3.65
Concentrating		2.11	1.67	2.56
Anxiety or depression		5.36	4.72	5.99
OCHS-EBS subscales, 2023	Range	Mean	95% CI	
ADHD	0–12	2.63	2.55	2.71
Internalizing symptoms	0–16	1.82	1.75	1.89
ODD	0–12	2.99	2.91	3.07

^1^ Due to the small number of children whose parents reported their gender as diverse, these cases were excluded from analyses.

Table 2 presents the results of regression analyses predicting children’s socio-emotional outcomes from their child care participation (unadjusted for covariates). Children who participated in child care in 2019 were less likely to have a functional difficulty making friends and tended to have higher scores on ADHD, internalizing, and ODD subscales in 2023.

The results of unadjusted regression models predicting socio-emotional functioning from the type of child care suggested that all types of care were associated with higher ADHD scores, whereas before- or after-school care was associated with more internalizing symptoms (Table 3). Conversely, participating in home- or centre-based care was associated with a lower likelihood of having a functional difficulty making friends. Since the crude associations between functional difficulty concentrating and functional difficulty with anxiety or depression and later socio-emotional outcomes were not significant, associations with these two predictors were not explored in subsequent models.

Results for models adjusted for socio-demographic variables, low positive parenting, and COVID-related learning disruptions are presented in Table 4. An examination of variance inflation factors suggested no concern for multicollinearity, with values below 1.25 for all variables. Similarly to the unadjusted models, all types of care were associated with higher ADHD scores, whereas centre-based care was associated with higher internalizing scores four years later. In the adjusted model, there was no association between any type of care and difficulty making friends. No significant interactions between positive parenting and child care participation or type were noted. However, in the model predicting internalizing symptoms, a significant interaction between low family income and child care type was found (F = 3.11, *p* = 0.026), suggesting that associations between child care type and internalizing symptoms differed depending on family income (see below).

Models stratified by low income are presented in Table 5. Among children living in low-income families, no associations were noted between any type of child care and internalizing symptoms. Among those living above the low-income threshold, participation in centre-based child care was associated with higher scores on the internalizing subscale four years later.

Results for the time spent in child care (Table 6) suggested that child care participation for 35 h or more was associated with a lower likelihood of a functional difficulty making friends, and participation for any amount of time was associated with higher ADHD scores. For internalizing and ODD symptoms, some differences were noted according to the time spent in child care; however, these effects were superseded by significant interactions with low family income and gender (see below).

Significant interactions were found between low family income and time spent in child care in models predicting internalizing (F = 4.86, *p* = 0.008) and ODD scores (F = 3.84, *p* = 0.023). In the model predicting ODD scores, an interaction between time in child care and child gender was additionally noted (F = 3.49, *p* = 0.031). The three-way interaction between time in child care, gender, and low family income was non-significant (F = 2.26, *p* = 0.105). Stratified models (Table 7) suggest that among children living in low-income families, participation in child care for 35 or more hours per week was associated with lower internalizing scores compared with no participation. Among those living above the low-income threshold, any degree of child care participation was associated with higher internalizing scores, and 35 h or more of child care was associated with higher ODD scores. Models stratified by child gender indicate that participation in child care for 35 h or more per week was associated with higher ODD scores among boys but not girls.

## 4. Discussion

This nationally representative longitudinal study suggests that child care participation among 1–5-year-olds is associated with socio-emotional functioning four years later and that these associations vary significantly according to child care type, time spent in care, low family income, and child gender. Broadly, participation in child care in early childhood (ages 1–5) was associated with higher ADHD scores at school age (ages 5–9). Child care attendance was associated with a lower likelihood of difficulty making friends but higher scores on measures of internalizing problems and ODD; however, important differences in these associations were noted according to the contextual factors examined, including the type of child care, time in child care, and household income.

First, results suggested meaningful differences in these associations based on the type of child care children attended. Compared with not attending child care, participation in centre-based care was associated with greater internalizing and ODD symptoms among those living above the low-income threshold. These findings, along with those suggesting that all types of child care were associated with ADHD symptoms, are largely in line with research suggesting that compared with other types of non-parental child care, centre-based care is associated with greater behaviour problems [9,11,33,34,35], an association that has been attributed to the higher ratio of children to caregivers in centre-based care [9]. Some research has suggested a fade-out effect, whereby associations between child care participation and behaviour problems are no longer significant by the time the child reaches the later elementary grades [12], a hypothesis we were unable to test in the present study due to the age range of children in our sample.

Second, though results were not consistent across all outcomes, this study yielded some evidence that associations between child care participation and socio-emotional outcomes were stronger when children spent longer hours in child care on a weekly basis. For example, compared to no time in child care in the reference week, attending child care 35 or more hours per week was associated with greater ODD symptoms among children living above the low-income threshold and lower internalizing symptoms among children living below this threshold. Child care attendance fewer than 35 h per week was not associated with these outcomes. Similarly, attendance for 35 h per week or more was associated with a lower likelihood of difficulty making friends. Associations with ADHD scores were similar regardless of the time spent in care. Some previous research has found that greater time spent in non-parental (and particularly centre-based) child care has been associated with more behaviour problems [36]; however, research employing a within-person design (that is, examining the impact of an increase or decrease in time spent in child care on children’s outcomes) has found negligible associations between time in centre-based care and externalizing behaviour [37].

As touched on above, associations between child care and internalizing and ODD symptoms differed according to family low-income status—among children living below the low-income threshold, child care participation for 35 h or more per week was associated with lower internalizing scores, and child care participation was not associated with ODD scores. Among children living above this threshold, centre-based child care was associated with higher internalizing scores. Much previous work has been devoted to the potential benefit of non-parental child care for children living in low-income families, particularly for children’s cognitive and linguistic development [1,23] but also their socio-emotional development, with some studies suggesting that child care attendance in early childhood is associated with fewer behaviour problems among low-income children into middle childhood [25]. The present study demonstrated some associations with positive functioning for children living in low-income households four years later with respect to fewer internalizing symptoms (spending more than 35 h a week in child care was associated with lower internalizing scores) and less difficulty making friends (this association was noted for all children, regardless of income). Although child care participation among low-income children also showed a trend towards lower ODD scores, these associations were not statistically significant.

In contrast, for children living above the low-income threshold, centre-based care was positively associated with internalizing symptoms, and participation in child care for 35 h or more per week was associated with higher ODD scores. There was some evidence that regardless of family income status the association with ODD was only significant among boys; however, the three-way interaction between gender, low income, and time in child care was not significant. A recent meta-analysis comparing the effects of universal child care for children from different socio-economic backgrounds suggested that positive outcomes of universal child care are noted more often for children of lower socio-economic status (SES), whereas negative effects are more often noted among children from higher SES backgrounds [38]. The present results add to this body of literature, suggesting that a ‘one-size-fits-all’ approach to child care may not be the most beneficial for children’s development [39,40,41].

Of note, means for all OCHS-EBS subscales were quite low, and associations between child care participation and functional difficulties in the internalizing domain (i.e., daily anxiety and depression) were not significant, suggesting that although child care was associated with elevated levels of these symptoms in higher-income children, it is possible that few (if any) children experienced clinically meaningful symptom levels.

In adjusted models, COVID-related educational disruptions and difficulties with online learning were both positively associated with difficulty making friends, internalizing symptoms, ADHD, and ODD (see Appendix A Table A1, Table A2, Table A3 and Table A4), suggesting that these unique experiences are indeed important to consider when assessing associations between children’s early education and later socio-emotional development. However, after adjusting for these experiences, associations between child care variables and outcomes largely remained significant. Moreover, no significant interactions between child care participation and COVID-related learning experiences were noted.

The seminal research on socio-emotional outcomes of child care by the NICHD suggests that parental and non-parental (e.g., by child care providers) caregiving both independently contribute to children’s social functioning [42], and some research suggests that the association between child care participation and children’s behaviour problems is more pronounced among children with parents scoring lower on sensitivity [1]. In the present study, though positive parenting was generally associated with fewer socio-emotional symptoms (see Appendix A Table A1, Table A2, Table A3 and Table A4), no significant interactions between child care participation and parenting were noted. These results seem to suggest that associations between child care participation and outcomes are independent of parenting behaviour; however, further research on other potential moderators and mediators related to parenting is warranted.

### Limitations and Future Directions

Though the present study has several notable strengths, including a longitudinal design and a relatively large sample size, there are nonetheless several limitations to be noted.

First, the measure of early childhood child care participation was collected at a single time point. It is possible that families’ use of child care changed during the study period, a factor we were not able to assess with the available data.

Second, our analyses examined children’s experiences of education disruptions due to COVID-19, which was particularly important to consider given the timeframe of the survey—with the first wave of data collected before the beginning of the pandemic and the second wave in the later stages, well after widespread lockdowns and public health measures had been lifted. As these disruptions occurred after the first wave of data collection, it is possible that they lie on the causal pathway between child care participation and socio-emotional outcomes (as child care participation may have been associated with the type and intensity of learning disruptions). Relatedly, although these COVID-related school disruptions are presumed to have taken place prior to the 2023 survey collection, it is possible that the socio-emotional outcomes assessed in 2023 had an onset prior to these disruptions. Therefore, the direction of effect could not be ascertained—it is plausible, for example, that children with higher symptom levels at baseline were more likely to struggle with their schooling during the pandemic. To mitigate this possibility, final models included parent-rated mental health at baseline, as this was the only measure of mental health or socio-emotional functioning available for children under 5 years old. Associations between child care participation and socio-emotional functioning remained significant after adjustment, lending confidence in our findings.

Third, though we examined the type of child care and time spent in care independently, it is possible that these factors interact—that is, that associations between child care type and outcomes differ by time spent in child care. However, sample size limitations precluded analyses of these interactions, as numbers of children in certain combinations of child care type and time were too small.

Last, while this study included several important contextual factors related to child care participation, including the type of care and time spent in care, like many previous studies we were unable to assess the quality of child care. Child care quality has been highlighted as an important characteristic to consider, with the potential to dampen and even reverse potential negative effects of time spent in child care on children’s behavioural outcomes [34,36]. Studies examining child care quality in tandem with other contextual factors are crucial to understanding the potential impacts of child care participation on children’s development.

## 5. Conclusions

Results of the present study highlight important contextual differences in associations between child care participation in early childhood and socio-emotional outcomes four years later. Future research examining the differential impacts of various types of child care, as well as outcomes for children from different socio-economic backgrounds—particularly research incorporating measures of child care quality—is crucial to the development of national child care strategies that offer advantages to all children.

## Figures and Tables

**Table 2 children-12-01463-t002:** Results of unadjusted logistic and linear regression predicting socio-emotional outcomes in 2023 from child care participation in 2019 (reference category: no regular child care).

Functional Difficulty	OR		95% CI		
Making friends	0.59	*	0.43	0.81	
Depression or anxiety	1.00		0.78	1.28	
Concentrating	1.18		0.77	1.80	
OCHS-EBS subscale scores	β		95% CI		*p* value
ADHD	0.304	*	0.146	0.462	<0.001
Internalizing	0.160	*	0.018	0.303	0.027
ODD	0.171	*	0.016	0.326	0.030

* estimate is statistically significant (*p* < 0.05).

**Table 3 children-12-01463-t003:** Results of unadjusted logistic and linear regressions predicting socio-emotional outcomes in 2023 from child care type in 2019 (reference category: no regular child care).

	Home-Based Care				Centre-Based Care				Before- or After-School Care			
Functional difficulty	OR	95% CI			OR	95% CI			OR	95% CI		
Making friends	0.55 *	0.37	0.83		0.63 *	0.43	0.92		0.50	0.24	1.03	
OCHS-EBS subscale scores	β	95% CI		*p* value	β	95% CI		*p* value	β	95% CI		*p* value
ADHD	0.247 *	0.040	0.453	0.019	0.308 *	0.120	0.496	0.001	0.465 *	0.132	0.798	0.006
Internalizing	0.000	−0.185	0.186	0.997	0.144	−0.018	0.305	0.082	0.668 *	0.341	0.995	<0.001
ODD	0.176	−0.028	0.380	0.091	0.137	−0.047	0.321	0.144	0.281	−0.051	0.614	0.097

* estimate is statistically significant (*p* < 0.05).

**Table 4 children-12-01463-t004:** Results of logistic and linear regressions predicting socio-emotional outcomes in 2023 from child care type in 2019, adjusted for socio-demographic factors, general mental health at baseline, low positive parenting, and COVID-related learning disruptions.

	Home-Based Care				Centre-Based Care				Before- or After-School Care			
Functional difficulty	OR	95% CI			OR	95% CI			OR	95% CI		
Making friends	0.65	0.41	1.04		0.75	0.49	1.14		0.50	0.21	1.15	
OCHS-EBS subscale scores	β	95% CI		*p* value	β	95% CI		*p* value	β	95% CI		*p* value
ADHD	0.265 *	0.060	0.470	0.012	0.283 *	0.086	0.481	0.005	0.488 *	0.160	0.816	0.004
Internalizing	0.115	−0.077	0.306	0.240	0.224 *	0.055	0.393	0.010	0.045	−0.308	0.397	0.803
ODD	0.173	−0.030	0.376	0.095	0.123	−0.066	0.312	0.203	0.094	−0.253	0.440	0.596

* estimate is statistically significant (*p* < 0.05). Note: models are adjusted for child age, child gender, general mental health at baseline, low family income, single-parent status in 2019, marital status change from 2019 to 2023, parent education, low positive parenting, difficulties with online learning during the COVID-19 pandemic, and learning disruptions due to the COVID-19 pandemic. The full models can be found in the Appendix A Table A1, Table A2, Table A3 and Table A4.

**Table 5 children-12-01463-t005:** Results of fully adjusted logistic regression predicting socio-emotional outcomes in 2023 from child care type in 2019, stratified by household low-income status (reference: no child care).

	Low-Income Family				Above Low-Income Measure Threshold			
	β	95% CI		*p* value	β	95% CI		*p* value
Internalizing problems								
Home-based care	0.131	−0.348	0.610	0.590	0.142	−0.060	0.344	0.169
Centre-based care	−0.157	−0.505	0.191	0.378	0.362 *	0.166	0.559	<0.001
Before- or after-school care	−0.561	−1.214	0.092	0.092	0.258	−0.144	0.660	0.208

* estimate is statistically significant (*p* < 0.05). Note: models are adjusted for child age, child gender, general mental health at baseline, single-parent status in 2019, marital status change from 2019 to 2023, parent education, low positive parenting, difficulties with online learning during the COVID-19 pandemic, and learning disruptions due to the COVID-19 pandemic.

**Table 6 children-12-01463-t006:** Results of fully adjusted logistic regression predicting socio-emotional outcomes in 2023 from amount of time in child care in the reference week, 2019 (reference: 0 h in child care).

	<35 h/Week				35 h or More/Week			
Functional difficulty	OR	95% CI			OR	95% CI		
Making friends	0.69	0.46	1.05		0.59 *	0.37	0.93	
OCHS-EBS subscale scores	β	95% CI		*p* value	β	95% CI		*p* value
ADHD	0.232 *	0.046	0.419	0.015	0.339 *	0.139	0.538	0.001
Internalizing	0.214 *	0.040	0.389	0.016	0.096	−0.075	0.268	0.270
ODD	0.101	−0.083	0.286	0.282	0.245 *	0.055	0.434	0.012

* estimate is statistically significant (*p* < 0.05). Note: models are adjusted for child age, child gender, general mental health at baseline, low family income, single-parent status in 2019, marital status change from 2019 to 2023, parent education, low positive parenting, difficulties with online learning during the COVID-19 pandemic, and learning disruptions due to the COVID-19 pandemic.

**Table 7 children-12-01463-t007:** Results of fully adjusted logistic regression predicting socio-emotional outcomes in 2023 from amount of time in child care in the reference week in 2019, stratified by household low-income status and child gender (reference: 0 h in child care).

	Low-Income Family				Above Low-Income Measure Threshold			
	β	95% CI		*p* value	β	95% CI		*p* value
Internalizing problems								
<35 h/week	0.074	−0.314	0.461	0.709	0.284 *	0.094	0.475	0.004
35 h or more/week	−0.412 *	−0.750	−0.073	0.017	0.247 *	0.051	0.443	0.013
ODD								
<35 h/week	−0.023	−0.410	0.363	0.906	0.165	−0.046	0.376	0.126
35 h or more/ week	−0.289	−0.728	0.149	0.196	0.398 *	0.183	0.613	<0.001
	Boys				Girls			
	β	95% CI		*p* value	β	95% CI		*p* value
ODD								
<35 h/week	0.205	−0.064	0.475	0.135	−0.020	−0.271	0.231	0.878
35 h or more/week	0.459 *	0.185	0.733	0.001	0.001	−0.260	0.263	0.991

* estimate is statistically significant (*p* < 0.05). Note: models are adjusted for child age, child gender, general mental health at baseline, low family income, single-parent status in 2019, marital status change from 2019 to 2023, parent education, low positive parenting, difficulties with online learning during the COVID-19 pandemic, and learning disruptions due to the COVID-19 pandemic.

## Data Availability

Data from the Canadian Health Survey on Children and Youth (CHSCY) are held by Statistics Canada and available for use by researchers though the Research Data Centres programme. Interested parties can receive more information here https://www.statcan.gc.ca/en/microdata/data-centres (accessed on 23 October 2025).

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
