# Peer review of "Socio-Emotional Outcomes of Child Care Participation: Results of a Four-Year Longitudinal Cohort Study"

_children, 2025, doi:10.3390/children12111463_

Round 1
Reviewer 1 Report
Comments and Suggestions for Authors
Thank you for the opportunity to review the manuscript. I think the authors have done a detailed and thorough analysis and clearly presented their findings and importance of the research.
However, I would recommend the following changes before publication:
1) Reference to work (introduction and discussion) that has followed children through different amounts of care and linked this to changes in outcomes. This existing work uses a within-person design and therefore removes time-invariant confounding. Which, unfortunately, we cannot do with between-person comparison due to the limitations of what we can measure.
Rey‐Guerra, C., Zachrisson, H. D., Dearing, E., Berry, D., Kuger, S., Burchinal, M. R., ... & Côté, S. M. (2023). Do more hours in center‐based care cause more externalizing problems? A cross‐national replication study. Child Development, 94(2), 458-477.
2) It is important to adjust for socio-demographic confounding and the authors have included many relevant covariates and noted some key missing ones (e.g., quality of care). But, development of the child before/at baseline of the intervention (childcare) is the most important covariate and is not included. Further, it is my understanding that the CHSCY 2019 does have measures (even if approximations) of children's development. Therefore measures of children's baseline development would be needed in the models. I would offer specifics but I am not familiar with all the available variables.
3) The authors adjust for child age in the models. But, child age is not reported in Table 1. I would say Table 1 needs to expanded by relevant age groups (e.g., 1, 2, 3, 4, 5 or some combination based on sample size). This would likely be best suited to supplementary material due to size. It may also be that if more information is available for older children, a stratified analysis by child age would be relevant.
4) I was unsure on the amount of missing data in 2019. If there was no missing data, it would be great to have this reported. If there was missing data, it would be great to have the amount reported for each variable in Table 1 and the authors might consider multiple imputation to increase their sample size, given the regression models are not too onerous to fit (I'm guessing based on sample size).
5) The summary of regression models is useful and relevant. But, it is insufficient to evaluate the models. I would suggest including at least one form of the models in supplementary material so it can be checked, perhaps the models from Table 4.
6) Including covid-related learning disruptions does seem relevant. But, as this occurred after the childcare attendance (2019) it is an intermediate outcome and therefore would be excluded from the models unless using a method that can handle intermediate outcomes (e.g., marginal structural models). However, I am aware common-practice and the realities of available data mean we sometimes include intermediate outcomes anyway. Therefore, I think a noted limitation and justification for why this is included could also suffice.
Overall, I think this an important contribution and commend the authors. The inclusion of baseline outcomes in the model would really strengthen the paper and catch the interest of many researchers, myself included. Additional details on the models and children's age would also help understand the validity and generalisability of the findings.
Author Response
1) Reference to work (introduction and discussion) that has followed children through different amounts of care and linked this to changes in outcomes. This existing work uses a within-person design and therefore removes time-invariant confounding. Which, unfortunately, we cannot do with between-person comparison due to the limitations of what we can measure.
Rey‐Guerra, C., Zachrisson, H. D., Dearing, E., Berry, D., Kuger, S., Burchinal, M. R., ... & Côté, S. M. (2023). Do more hours in center‐based care cause more externalizing problems? A cross‐national replication study. Child Development, 94(2), 458-477.
RESPONSE: Thank you for this suggestion, we have cited the suggested article in the revised manuscript (p.12).
2) It is important to adjust for socio-demographic confounding and the authors have included many relevant covariates and noted some key missing ones (e.g., quality of care). But, development of the child before/at baseline of the intervention (childcare) is the most important covariate and is not included. Further, it is my understanding that the CHSCY 2019 does have measures (even if approximations) of children's development. Therefore measures of children's baseline development would be needed in the models. I would offer specifics but I am not familiar with all the available variables.
RESPONSE: We agree that adjusting for socio-emotional adjustment at baseline would be ideal. Unfortunately, availability of measures for children under 5 years old is extremely limited. On your advice, we have included parents’ reports of children’s general mental health (the only mental health indicator collected for this young age group) in the adjusted models.
3) The authors adjust for child age in the models. But, child age is not reported in Table 1. I would say Table 1 needs to expanded by relevant age groups (e.g., 1, 2, 3, 4, 5 or some combination based on sample size). This would likely be best suited to supplementary material due to size. It may also be that if more information is available for older children, a stratified analysis by child age would be relevant.
RESPONSE: We report child age in our revised table 1.
4) I was unsure on the amount of missing data in 2019. If there was no missing data, it would be great to have this reported. If there was missing data, it would be great to have the amount reported for each variable in Table 1 and the authors might consider multiple imputation to increase their sample size, given the regression models are not too onerous to fit (I'm guessing based on sample size).
RESPONSE: Thank you for this suggestion- we analysed percentage missing at time 1 as you suggest and mention this in the revised method section. As the percentage was low (<1% for all variables) we do not perform imputation.
5) The summary of regression models is useful and relevant. But, it is insufficient to evaluate the models. I would suggest including at least one form of the models in supplementary material so it can be checked, perhaps the models from Table 4.
RESPONSE: We are unfortunately unsure what the reviewer means by the form of the model. We have included a regression summary including all covariates from Table 4 in the Appendix, supplementary tables S1-S4.
6) Including covid-related learning disruptions does seem relevant. But, as this occurred after the childcare attendance (2019) it is an intermediate outcome and therefore would be excluded from the models unless using a method that can handle intermediate outcomes (e.g., marginal structural models). However, I am aware common-practice and the realities of available data mean we sometimes include intermediate outcomes anyway. Therefore, I think a noted limitation and justification for why this is included could also suffice.
RESPONSE: thank you for this suggestion, we have include this limitation in our revised manuscript.
Reviewer 2 Report
Comments and Suggestions for Authors
I really enjoyed reading this paper, it is well researched, has a good sample size, and a clear methodology. The findings are suitably qualified and justified. I think I would have liked to have known more about 'quality' in early childcare in Canada which would have provided more context for my understanding as childcare systems can differ. There are some hints at quality e.g. l.368 where adult:child ratios are mentioned - although this could have been more explicitly linked. However, I appreciate that the paper is providing a big picture so may be out of the overall scope of the author's intentions. On that assumption I would say this is publishable in it's current form (except ll166-168 there is a ' missing from each reference to 'a few times a year'
Author Response
I think I would have liked to have known more about 'quality' in early childcare in Canada which would have provided more context for my understanding as childcare systems can differ. There are some hints at quality e.g. l.368 where adult:child ratios are mentioned - although this could have been more explicitly linked. However, I appreciate that the paper is providing a big picture so may be out of the overall scope of the author's intentions. On that assumption I would say this is publishable in it's current form (except ll166-168 there is a ' missing from each reference to 'a few times a year'
RESPONSE: Thank you for catching this typo – we have fixed it. With regards to quality, we agree it is an important aspect of child care to consider, as mentioned in our discussion. As we were not able to assess quality in the present study we refrain from a deeper discussion of this aspect of child care.
Reviewer 3 Report
Comments and Suggestions for Authors
The current study investigates associations between child care participation (ages 1–5) and socio-emotional outcomes four years late. Main findings show child care linked to better friend-making skills but higher symptoms of ADHD, ODD, and internalizing problems, with effects varying by care type (e.g., centre-based strongest for ODD/internalizing) and family income. Associations persisted after adjusting for confounders like parenting quality and COVID-19 school disruptions. It is an important study which will contribute to literature.
- The title of the article should be more specific and reflect the content of the study.
- Aims of the study are clearly depicted. The hypothesis of the study should also be added.
- Retention rate (54.2%) for the waves of the cohort study is quite low for longitudinal claims. An attrition analysis should have been done. This problem should be mentioned in limitations section.
- Catgeorization of child care as yes or no is very simplistic but acceptable for such large study. This approach’s limitation should be mentioned as it widely ignores duration of intensity of care during five year period.
- Clarify the psychometric properties of measures used especially positive parenting scale
- More details are needed in analysis section. Describe how you handle multicollinearity and correction for multiple comparisons. Power analysis findings should also be added.
- Discussion is overall satisfactory.
Author Response
- The title of the article should be more specific and reflect the content of the study.
RESPONSE: While we respect this opinion, we would prefer not to change the title of the paper.
2. Aims of the study are clearly depicted. The hypothesis of the study should also be added.
RESPONSE: We prefer not to state hypotheses post-hoc, but are happy to add these if the journal requires this.
3. Retention rate (54.2%) for the waves of the cohort study is quite low for longitudinal claims. An attrition analysis should have been done. This problem should be mentioned in limitations section.
RESPONSE: Thank you for this suggestion; we have conducted an attrition analysis and mentioned it in text. Those who did not respond in 2023 were less likely to use child care, more likely to live below the low income threshold, and less likely to have very good or excellent mental health at baseline. Of note, the longitudinal survey weights are designed to adjust for nonresponse to provide more unbiased estimates.
4. Catgeorization of child care as yes or no is very simplistic but acceptable for such large study. This approach’s limitation should be mentioned as it widely ignores duration of intensity of care during five year period.
RESPONSE: we agree- we have added to the limitations section as you suggest.
5. Clarify the psychometric properties of measures used especially positive parenting scale
RESPONSE: We have added additional information on scales as you suggest.
6. More details are needed in analysis section. Describe how you handle multicollinearity and correction for multiple comparisons. Power analysis findings should also be added.
RESPONSE: We have added to the manuscript regarding tests for multicollinearity. VIFs ranged from 1.08 – 1.25, therefore no concern was raised for multicollinearity.
We prefer not to conduct post-hoc power analysis as recommended here: The Interpretation of Statistical Power after the Data have been Gathered - PMC and instead include confidence intervals for coefficients as an alternative, as suggested by the authors of the linked article.
As our selection of variables for inclusion in the regression models is theoretically based, and we focus on coefficients for child care variables only, we do not conduct any correction for multiple comparisons here.
7. Discussion is overall satisfactory.